# Evidence-Based Approach for Secondary Prevention of Uterine Fibroids (The ESCAPE Approach)

**DOI:** 10.3390/ijms242115972

**Published:** 2023-11-04

**Authors:** Somayeh Vafaei, Michał Ciebiera, Mervat M. Omran, Mohammad Mousaei Ghasroldasht, Qiwei Yang, Tanya Leake, Rochelle Wolfe, Mohamed Ali, Ayman Al-Hendy

**Affiliations:** 1Department of Obstetrics and Gynecology, University of Chicago, Chicago, IL 60637, USA; somayehv@bsd.uchicago.edu (S.V.); mervatomran@bsd.uchicago.edu (M.M.O.); mmghasroldasht@bsd.uchicago.edu (M.M.G.); yangq@bsd.uchicago.edu (Q.Y.); 2Second Department of Obstetrics and Gynecology, Center of Postgraduate Medical Education, 00-189 Warsaw, Poland; michal.ciebiera@gmail.com; 3Warsaw Institute of Women’s Health, 00-189 Warsaw, Poland; 4Development and Research Center of Non-Invasive Therapies, Pro-Familia Hospital, 35-302 Rzeszow, Poland; 5The White Dress Project, Atlanta, GA 30309, USA; tanya@thewhitedressproject.org (T.L.); rochelle@thewhitedressproject.org (R.W.)

**Keywords:** uterine fibroid, leiomyoma, secondary prevention, vitamin D3, EGCG, reproductive health

## Abstract

Uterine fibroids (UFs) are common tumors in women of reproductive age. It is imperative to comprehend UFs’ associated risk factors to facilitate early detection and prevention. Simple relying on surgical/pharmacological treatment of advanced disease is not only highly expensive, but it also deprives patients of good quality of life (QOL). Unfortunately, even if the disease is discovered early, no medical intervention is traditionally initiated until the disease burden becomes high, and only then is surgical intervention performed. Furthermore, after myomectomy, the recurrence rate of UFs is extremely high with the need for additional surgeries and other interventions. This confused approach is invasive and extremely costly with an overall negative impact on women’s health. Secondary prevention is the management of early disease to slow down its progression or even halt it completely. The current approach of watchful observation for early disease is considered a major missed opportunity in the literature. The aim of this article is to present an approach named the ESCAPE (Evidence-Based Approach for Secondary Prevention) of UF management. It comprises simple, inexpensive, and safe steps that can arrest the development of UFs, promote overall reproductive health, decrease the number of unnecessary surgeries, and save billions of health care systems’ dollars worldwide.

## 1. Introduction

Uterine fibroids (UFs), also known as leiomyomas or myomas, represent a significant global health issue, impacting a substantial portion of reproductive age women, with estimates suggesting that they affect even up to 80% of women worldwide [1,2]. The burden of UFs on a global scale has markedly increased, with a rise in both incidences and disability-adjusted life years [3,4]. This increase is particularly prominent in regions with a lower sociodemographic index, indicating a pressing need for immediate attention to be given to health protection of underprivileged communities [5]. Unfortunately, UFs are often accompanied by distressing symptoms, which can significantly impact the well-being of those affected. These symptoms include heavy menstrual bleeding (HMB) leading to anemia [6], painful periods, pelvic pain or pressure, abdominal swelling or bloating, and discomfort during sexual intercourse, frequent urination, constipation, back or leg pain, infertility, or miscarriage [7]. The presence of these symptoms underscores the importance of addressing UFs and finding effective management strategies to improve the quality of life (QOL) for individuals experiencing them [8,9]. 

Although the etiology of UFs is not fully understood, several risk factors, including age, race, obesity, parity, hypertension, vitamin D deficiency, hormonal imbalances, exposure to endocrine-disrupting chemicals (EDCs) (organophosphate esters and plasticizers) [10,11], genetic polymorphism, and lifestyles involving a worse diet, lack of physical activity, and high stress have been identified [12,13]. Furthermore, some of the variations in UFs’ risk and the racial disparities associated with UF traits can be attributed to genetic differences among various geographic populations. UFs disproportionately affect black women, who not only experience UFs at an earlier age but also face a higher overall risk [14,15]. Given these factors, even seemingly healthy individuals should be mindful of preventive measures. Our prior and ongoing research has contributed to the development of a groundbreaking mechanistic paradigm that provides insight into the development of UFs.

Primary prevention involves measures to prevent disease in healthy individuals by either reducing risk exposure or enhancing immunity. It aims to halt the disease’s development in high risk before it even starts [16]. Currently, there is no well-established screening test for UFs and, as such, no viable approach for primary prevention, However, our group and others are pursuing this important goal [17]. UFs are diagnosed mostly by bimanual pelvic test, especially in advanced disease in thin patients. Imaging usually using ultrasonography and occasionally using magnetic resonance imaging (MRI), or computed tomography (CT) is the most common method to diagnose UF [2,18,19]. Imaging allows proper evaluation of the disease such as the size, location, and multiplicity of UFs. Two crucial aspects in contemporary approaches to optimal treatment include the concepts of shared decision making and the availability of personalized therapy options, which may include a “watchful waiting” approach determined by the patient’s symptoms, their individual treatment objectives, and the physician’s assessment. Medical treatment included non-hormonal and hormonal approaches. Non-hormonal like catechol-*O*-methyl transferase. (COMT) inhibitors, vitamin D [20,21], vitamin D receptors (VDR) agonists, epigallocatechin gallate (EGCG), localized gene therapy, nanoparticles, and localized bacterial collagenase [22,23,24,25,26]. Hormone therapy such as gonadotropin-releasing hormone (GnRH) agonists and antagonist, selective progesterone receptor modulators (SPRMs), aromatase inhibitors or progestin-releasing intrauterine devices (IUDs) [27,28]. Surgery consists of traditional (abdominal myomectomy and hysterectomy) or minimally invasive or noninvasive procedures (laparoscopic or robotic myomectomy and hysterectomy, uterine artery embolization, radiofrequency ablation or endometrial ablation, MRI or ultrasound guided focused ultrasound surgery (FUS), may sometimes be necessary to remove or shrink large or symptomatic UFs [29,30,31,32,33]. Annual direct and indirect UFs costs range from $5.9 to $34.4 billion for United States of America women seeking treatment. UFs complicated by HMB were associated with significantly higher direct health care costs compared with UFs or HMB alone [34,35]. 

Secondary prevention involves detecting early-stage diseases in individuals with subtle signs of illness. This phase is especially relevant for women concerned about UF recurrence post-surgery. Strategies are pursued to prevent disease advancement or recurrence [36]. In the context of secondary prevention, screenings are frequently employed as a mean to achieve early disease detection and intervention. Currently, there is no validated method for screening pre-symptomatic health women for risk of future UFs development [37,38]. Based on our team’s prior investigations, setting a screening, supplementation, treatment guidelines, and public health strategies for vitamin D deficiency in women with UFs as well as women at a high risk of UF development might be of potential importance [39]. Recent investigations into the use of shear wave elastography (SWE) as a potential screening tool have indicated its potential utility [17]. We advocate the opinion that ultrasound elastography (USE), with a particular focus on SWE, has the potential to enhance the evaluation of tissue stiffness. This improvement in assessing tissue stiffness can have a significant impact on the diagnosis and treatment of conditions such as adenomyosis, fibroids, endometrial lesions, cervical cancer, as well as the precise management of preterm birth and monitoring during intrauterine insemination. This innovative approach allows for the detection of subtle changes in the elasticity of pre-fibroid myometrial tissue, potentially serving as an effective method for identifying individuals who may be at risk of developing UFs in the future [40,41]. Additionally, we are investigating the possible utility of urinary inflammatory markers to either solely or in combination with SWE accurately identify pre-symptomatic women at risk of future development of UFs. Armed with this knowledge, one can envision the implementation of several preventive strategies that are both safe and fertility friendly. These strategies aim to prevent the development of UFs and can conceivably eradicate or dramatically decrease the prevalence of UFs worldwide [17]. We undertook a comprehensive search for studies across multiple databases, including PubMed, Scopus, Embase, and ISI Web of Science related to the early detection and prevention of uterine fibroids, aiming to help women with strategies to reduce their risk of developing uterine fibroids and improve their overall gynecological health.

Given the negative impact on QOL, the unknown exact cause of UFs, and high-cost limited treatment options, we propose the ESCAPE approach (Evidence-Based Approach to Secondary Prevention of UFs) for the prevention of this condition. The primary focus of ESCAPE is on early detection and effective management of UFs. By implementing these simplified protocols, individuals can reduce their risk of developing UFs and enhance their overall gynecological health [42,43]. Women can take control of their health and reduce the risks associated with UFs by following recommended guidelines. They should consult with their healthcare providers to tailor the approach to their individual needs. Collectively increased awareness of our ESCAPE approach can potentially improve afflicted women’s QOL, especially women of color who withstand the major burden of this serious disease. Additionally, gynecologists should be aware of these steps and communicate them to their patients, emphasizing preventive measures and regular check-ups.

## 2. Vitamin D Serum Level Checkup and Proper Supplementation

Vitamin D is an essential fat-soluble vitamin that plays a crucial role in various aspects of health, including maintaining calcium balance, supporting bone health, and modulating the immune system [44,45]. The presence of VDR in uterine tissue and the observed associations between vitamin D deficiency and various gynecological conditions, including UFs, have sparked interest in investigating the potential benefits of vitamin D in the management of UFs [46,47,48,49]. Encouraging screening, supplementation, treatment guidelines, and public health strategies for vitamin D deficiency in women with UFs as well as women at a high risk of UFs development might be important [39,50,51]. In vitro studies proved vitamin D efficacy in inhibiting UF growth by targeting pathways involved in the regulation of various biological processes, including proliferation, extra cellular matrix (ECM) remodeling, DNA repair, and apoptosis [22,52,53]. 

Vitamin D supplementation is commonly recommended in oral or injectable forms to address deficiencies and support overall health. The active form of vitamin D, known as 1,25-dihydroxyvitamin D3 (1,25(OH)_2_D_3_), plays a crucial role in maintaining the balance of calcium and phosphate in the body [54,55]. The process of activating vitamin D involves several steps. It begins with the synthesis of prohormone vitamin D in the skin, which is stimulated by exposure to sunlight. Subsequently, this precursor is converted to 1,25(OH)_2_D_3_ in the liver and kidneys. In addition to supplementation, dietary sources of vitamin D include fatty fish like tuna, mackerel, and salmon, as well as beef liver, egg yolks, and fortified foods such as dairy products, orange juice, and breakfast cereals [56,57,58]. Vitamin D deficiency can be linked to gut dysbiosis and altered estrogen metabolism that mandate further investigations as well as how these changes play key roles in the pathogenesis of UFs [59,60]. Notably, in 2010, our group identified vitamin D deficiency as a novel risk factor for UFs [52,61,62,63], a finding that was later confirmed by published reports globally [46]. In fact, lower serum vitamin C, vitamin D [64,65,66], and calcium levels occurred in women with UFs [67]. This observation can also explain the disproportionally high incidence of UFs among Black women, who are also ten times more likely to be vitamin D-deficient than White women [68,69,70,71]. Women with low parity, those belonging to lower socioeconomic status and those having less than 1 h sun exposure per day were independently found to have high risk for development of UFs [72,73]. In addition to 1,25-dihydroxyvitamin D3 (1,25(OH)_2_D_3_), paricalcitol, an analog of 1,25(OH)_2_D_3_ significantly reduced UF tumor size, and the shrinkage was slightly higher in the paricalcitol-treated group [74]. The role of vitamin D in the context of UF growth factors is closely tied to the activation of the Ras/Raf/MEK/ERK pathway, where the membrane-bound receptors for estradiol, progesterone, and vitamin D intersect. These receptors collaborate to initiate a cascade of events. Noteworthy elements involved in this interplay encompass various substances such as steroids, growth factors, transforming growth factor (TGF)-β and its downstream mediator Smad, wingless-type (Wnt) signaling including β-catenin, retinoic acid, vitamin D, and peroxisome proliferator-activated receptor γ (PPARγ). A significant observation is the convergence of multiple pathways, often synergistically. For instance, the mitogen-activated protein kinase (MAPK) and Akt pathways exhibit a role as signal integrators, harmonizing inputs from diverse signaling routes, which encompass growth factors, estrogen, and vitamin D [75]. In proteomics, 20 proteins mainly functioned in transportation (such as apolipoprotein A-I) and coagulation (such as the fibrinogen gamma chain) were identified. In the middle of these plasma markers of UFs, vitamin D-binding protein were considered important [76]. Nonsynonymous single-nucleotide polymorphisms (SNPs) that result in missense substitutions within tryptic peptides and are prevalent in European, African, and East Asian populations have been identified as peptide ancestry informative markers (pAIMs) through the analysis of the 1000 Genomes dataset. Among these candidates is a D451E substitution in the GC vitamin D-binding protein, a variant previously linked to altered vitamin D levels in both African and European populations [77]. Positive correlation between pretreatment values of serum 1,25(OH)_2_D_3_ as well as its decrease and the reduction in bone mass during GnRH agonist UF treatment were reported [78].

Vitamin D has shown promising anti-proliferative effects in UFs [79]. Studies have indicated that vitamin D can inhibit the proliferation of UFs cells and induce cell cycle arrest, preventing their uncontrolled growth [52,80]. Vitamin D inhibits the growth of UF cells through the downregulation of kinases and Bcl2 and suppresses COMT expression and activity [22]. Moreover, a vitamin D-deficient diet in mice triggered low serum vitamin D levels and resulted in increased expression of steroid receptors, increased expression of proliferation-related genes, and enhanced inflammation and DNA damage in the murine myometrium [81,82,83]. These anti-proliferative effects of vitamin D in UFs suggest its potential as a therapeutic target or supplement in the management of this condition [84,85,86]. Human UFs contain significantly lower 1,25(OH)_2_D_3_ than its adjacent at-risk myometrium (Myo-F). UFs, Myo-F, and normal myometrium (Myo-N) express CYP27B1 that codes for 1-α hydroxylase (vitamin D activating enzyme) and CYP24A1 that codes for 24-hydroxylase (vitamin D catabolizing enzyme). UFs express a significantly higher level of CYP24A1 than Myo-N, indicating that overexpression of 24-hydroxylase is a mechanism by which UFs sustain a relative state of vitamin D and plays a role in the induction of apoptosis, or programmed cell death, in UFs [87,88]. VDRs bind to specific DNA sequences, known as vitamin D response elements, and modulate the expression of genes related to apoptosis. By activating these genes, vitamin D promotes the death of UF cells and inhibits their growth [89]. 

Angiogenesis, the process of forming new blood vessels, plays a critical role in the growth and maintenance of UFs. When vitamin D administered as a treatment, it has been observed to decrease the expression of pro-angiogenic factors in UFs cells, thereby inhibiting the formation of new blood vessels [90,91]. This anti-angiogenic effect is significant as it may contribute to the reduction of UF size and vascularity [85]. Studies have suggested that vitamin D can inhibit the production of pro-angiogenic factors, such as vascular endothelial growth factor (VEGF), thereby reducing the formation of blood vessels within UFs tissue [92]. By targeting angiogenesis, vitamin D may disrupt the blood supply to the fibroid, impeding its growth and potentially promoting regression.

Vitamin D has been studied for its potential anti-inflammatory effects in UFs. Inflammation is a key component of UFs development and growth. Research suggests that vitamin D may regulate the expression of pro-inflammatory cytokines and enzymes involved in inflammatory processes. By modulating these inflammatory mediators, vitamin D may help reduce the inflammation associated with UFs [81].

Oxidative stress, caused by an imbalance between the production of reactive oxygen species (ROS) and the body’s ability to detoxify them, is believed to play a role in UF development and progression. Vitamin D has been shown to possess antioxidant properties, which can help neutralize ROS and reduce oxidative damage. By acting as an antioxidant, vitamin D may protect UF cells from oxidative stress and potentially inhibit their growth [93,94,95].

ECM is a network of proteins and other molecules that provide structural support to tissues. In fibroids, excessive deposition of ECM components, such as collagens and fibronectin, contributes to the growth and development of these tumors. Vitamin D has been found to interfere with the activities of enzymes involved in ECM remodeling, such as matrix metalloproteinases (MMPs) [96,97]. These mechanisms contribute to the degradation and reconstruction of the ECM. The vitamin D’s anti-fibrotic impact on UFs is additionally corroborated by its ability to decrease the expression of ECM proteins like fibronectin and collagen type 1 in UF cells, which are induced by TGFβ3 [98,99,100,101,102]. Additionally, there is a reduction in the expression of WNT4, β-catenin, and MMP9, all of which were found to be overexpressed in UFs [103,104]. Vitamin D functions as an inhibitor of Wnt4/β-catenin and mTOR signaling pathways, which play major roles in UF pathogenesis [105]. Higher BMI, positive family history, and lower vitamin D and higher TGF-β3 serum concentrations are risk factors for UFs [100].

However, clinical studies supported only in part the beneficial effects of vitamin D supplementation in reducing UF growth and tumor volume. Randomized controlled trials (RCT) and large population studies are mandatory as the potential clinical benefits are likely to be substantial to further explore its potential therapeutic applications [84,98,106]. Studies have indicated that vitamin D supplementation can lead to a decrease in UF size, providing a potential non-surgical approach to managing this condition [107]. These results were also supported in female Eker rats (14–16 months old) harboring UFs and showed that 1,25(OH)_2_D_3_ is an antitumor agent [108]. Notably, high concentrations of vitamin D can decrease UFs development but are limited by the participants with serum 25(OH)D ≥ 30 ng/mL [109]. Corachan et al. found that treatment with vitamin D did not change UF size when it was used for a short time; however, long-term usage induced a significant tumor volume reduction due to the lower cell proliferation rate in the human xenograft animal model [87]. A double-blinded clinical trial showed that the UF size in the vitamin D group was significantly decreased as compared to the placebo group [107]. Furthermore, another double-blinded trial reported the change in UF volume comparing a group of women who received 50,000 IU of vitamin D weekly for 12 weeks with whom received placebo [106]. A third, longer in duration, studied either the effect of vitamin D (1000 IU tablet; n = 55), or placebo (n = 54) daily for 12 months and concluded that UFs in the intervention group were significantly smaller in size than those in controls [110]. An additional study concluded that vitamin D supplementation is effective in reducing the progression to an extensive disease, and thus may alleviate the need of conventional surgical or medical therapy [65]. Importantly, in all these studies in several countries and a diverse population, vitamin D use was associated with an outstanding safety profile with no reported adverse events.

Hormonal imbalances, specifically estrogen dominance, are considered significant factors in the development of UFs. Vitamin D has emerged as a potential modulator of estrogen metabolism and signaling pathways [111,112]. Research conducted by Baird et al. revealed a compelling association between higher vitamin D levels and a reduced risk of UFs among premenopausal women [62]. This suggests that maintaining adequate vitamin D levels may play a role in gating the risk of UF development by influencing estrogen-related processes [52]. Estrogen receptors (ERs) play a crucial role in the growth and development of fibroids, as they respond to estrogen signals and contribute to the proliferative effects of this hormone [62]. By decreasing the expression of ERs, vitamin D supplementation may help modulate the responsiveness of UF tissue to estrogen, potentially inhibiting their growth and progression [113]. Our team observed that the expression of ER-α and progesterone receptors (PR-A and PR-B) showed an inverse correlation with VDR expression in UF tissue compared to Myo-N. Furthermore, the group noted that estrogen treatment decreased VDR expression while up-regulating PR-A and PR-B in UF cells [114]. These findings suggest that vitamin D acts as an antagonist to sex hormones in UF cells and may have potential as an anti-UF treatment [115,116]. A-kinase anchoring protein 13 (AKAP13) interacts with the VDR to alter vitamin D-dependent signaling in UF cells. AKAP13 inhibited the VDR activation by a mechanism that required, at least in part, RhoA activation [117]. In relation to the origin of UFs, specific genetic variations and polymorphisms have been linked to the VDR gene, which could conceivably function as mitigating factors lowering the risk of UFs development [118,119]. Studies have shown that vitamin D can exert immunomodulatory effects by enhancing the activity of regulatory T cells, reducing the production of pro-inflammatory cytokines, and promoting an anti-inflammatory environment. In the context of UFs, immune dysregulation has been implicated in their development and growth [120,121]. Vitamin D has been associated with anti-proliferative and anti-inflammatory effects, which are relevant to UF development and growth [122]. By optimizing vitamin D levels, it may be possible to create an environment that is less conducive to UF recurrence. Vitamin D has been investigated for its potential role in reducing the severity of specific symptoms associated with UFs [123]. VDR expression suggests a possible link between vitamin D levels and UF symptoms. Vitamin D deficiency is associated with more severe symptoms in women with UFs [124,125]. Based on networking discussions, the ranges of total serum 25-hydroxyvitamin D concentration indicating vitamin D deficiency (<20 ng/mL (<50 nmol/L)), suboptimal status (20–30 ng/mL (50–75 nmol/L)), and optimal concentration (30–50 ng/mL (75–125 nmol/L)) were confirmed [126,127]. Supplementation with vitamin D may help alleviate symptoms by modulating hormonal pathways, reducing inflammation, and influencing the growth of UF tissue. It is important to consult with a healthcare professional for personalized advice on incorporating vitamin D as part of a comprehensive treatment approach for managing UF symptoms [117,128].

By supplementing with optimized vitamin D’s diverse physiological functions, patients undergoing UF treatment may experience enhanced therapeutic outcomes, including reduced UFs size, alleviation of symptoms, and improved overall well-being, which was confirmed in different RCTs such as NCT03586947 and NCT03584529 [109,129]. Vitamin D has been investigated for its potential role in improving health related QOL outcomes in women with UFs with emphasis on the impact to a woman’s physical, emotional, and social well-being, leading to reduced QOL. Vitamin D deficiency has been associated with poorer QOL outcomes in various health conditions, and similar patterns have been observed in women with UFs [130]. Studies have reported associations between vitamin D deficiency and impaired fertility, including decreased pregnancy rates and increased risk of pregnancy complications [131,132]. It was reported that the predictors of UFs were fewer hours spent outdoors, middle social class and low vitamin D levels [133]. Vitamin D and its analogs seem to be promising, effective, and low-cost compounds in the management of UFs and their clinical symptoms, and the anti-tumor activities of vitamin D play an important role in UF biology. The synergy between vitamin D and selected anti-UF drugs is a very interesting issue which requires further research [46,134]. Optimizing vitamin D levels through supplementation or other means has the potential to improve QOL outcomes in women with UFs by supporting reproductive health (Figure 1).

## 3. Epigallocatechin Gallate (EGCG), and Green Tea Extract Consumption

Green tea is derived from the leaves of the Camellia sinensis plant and is rich in polyphenols, particularly catechins. These polyphenols have been extensively studied for their potential health benefits, including their anti-inflammatory, antioxidant, and anti-cancer properties [135,136]. As a natural product, EGCG has shown anti-proliferative, anti-angiogenic, anti-metastatic and pro-apoptotic abilities in reproductive diseases [137,138,139]. Green tea’s potential impact on UFs stems from these bioactive compounds. EGCG is the primary catechin subtype found in green tea. Some studies have suggested that drinking green tea may help prevent the development of UFs [24,136]. While in vitro and animal studies have shown promising results, the clinical data on the specific use of green tea in treating UFs are limited [140]. Observational studies exploring the relationship between green tea consumption and UF incidence or progression have yielded mixed results through various mechanisms. Based on one study, an observed safe level (OSL) of EGCG (704 mg/day) might be considered for tea preparations in beverage form based on human adverse events data. [141].

EGCG has been shown to inhibit the growth and proliferation of UF cells in both in vitro and animal studies. EGCG has been found to suppress the expression of proteins involved in cell proliferation and survival, such as cyclins and anti-apoptotic factors [142]. High dose of EGCG on rat ELT3 UF cells inhibited UF cell growth. Treatment with EGCG led to a 40% decrease in cell proliferation at 72 h along with significant reduction of proliferation markers expression such as proliferating cell nuclear antigen (PCNA) and cyclin-dependent kinases (CDKs)-4 proteins [143,144]. According to a study by Ahmed et al., the Pro-EGCG analogs 2a and 4a reduced susceptibility to COMT-mediated methylation and were found to inhibit proteasome and Akt signaling pathways [145]. EGCG significantly lowered the concentration of curcumin required to inhibit the AKT-mTOR pathway, reduce cell proliferation and induce apoptosis in uterine LMS cells by enhancing intracellular incorporation of curcumin [146].

EGCG has been investigated for its potential to induce apoptosis which plays a crucial role in maintaining tissue homeostasis by eliminating damaged or unwanted cells. EGCG effectively down-regulated CDK2 and CDK4, induced apoptosis, and blocked angiogenesis and MMPs [147,148]. Also, EGCG induced apoptosis with concentration-dependent downregulation of anti-apoptotic protein Bcl2 and upregulation of pro-apoptotic Bax [149]. Thus, EGCG can upregulate pro-apoptotic proteins while downregulating anti-apoptotic proteins, shifting the balance towards cell death. It can also induce mitochondrial dysfunction, leading to the release of cytochrome c and the activation of caspases, which are key mediators of apoptosis. Zhang et al. observed a 14-fold increase in bone morphogenetic protein 2 (BMP2) expression in UF culture treated with EGCG compared to untreated control [71,96,102]. BMP2 is a part of the TGF-β family and plays a vast role in cell growth, division, differentiation, and programmed cell death. Therefore, the ability of EGCG to increase BMP2 activity in tumors may lead to further investigating EGCG as an option for UF treatment [150,151].

EGCG suppressed the proliferation and migration of endothelial cells, thereby inhibiting angiogenesis. The anti-angiogenic effects may contribute to the reduction of UF size and vascularity by suppressing the production of pro-angiogenic factors, such as VEGF and MMPs [152,153]. EGCG has been shown to inhibit the production and activity of pro-inflammatory molecules, such as cytokines, chemokines, and inflammatory enzymes. By modulating these inflammatory mediators, EGCG can help reduce the inflammatory response associated with UF growth. Furthermore, EGCG has been found to suppress the activation of nuclear factor-kappa B (NF-κB), a key regulator of inflammation, in UF cells [154]. NF-κB activation triggers the expression of genes involved in inflammation and cell proliferation. By inhibiting NF-κB, EGCG can help mitigate the inflammatory processes driving UF development [155].

Oxidative stress, caused by an imbalance between the production of ROS and the body’s antioxidant defenses, is implicated in UF development and progression [156]. EGCG acts as a free radical scavenger and can neutralize ROS, thereby reducing oxidative stress. This antioxidant activity helps protect cells from oxidative damage and maintain their normal function [157]. EGCG also promotes the activity of endogenous antioxidant enzymes, such as superoxide dismutase (SOD) and catalase, which further contribute to the elimination of ROS [158]. By reducing oxidative stress, EGCG may help mitigate the pathological processes associated with UF growth and potentially prevent their occurrence.

Studies have indicated that EGCG can inhibit the synthesis and secretion of ECM proteins by UF cells. Additionally, EGCG has been found to interfere with the activities of enzymes involved in ECM remodeling, such as MMPs, which are responsible for breaking down and rebuilding the ECM [159]. EGCG treatment significantly reduced mRNA or protein levels of key fibrotic proteins, including fibronectin, collagen type I (COL1A1), plasminogen activator inhibitor-1 (PAI-1), connective tissue growth factor (CTGF), and actin alpha 2, smooth muscle (ACTA2) in UF cells, suggesting antifibrotic effects. EGCG treatment altered the activation of YAP, β-catenin, JNK and AKT, but not Smad 2/3 signaling pathways involved in mediating fibrotic process [160]. EGCG has been investigated for its potential role in the shrinkage of UFs volume. Although limited and inconclusive, some laboratory studies and animal models have shown promising results, suggesting that EGCG may possess anti-UF properties by inhibiting the growth of UF cells or inducing their apoptosis. However, the translation of these findings into human studies is still in its early stages [24]. While a small pilot study demonstrated a reduction in UF volume and symptom improvement in some participants following the administration of EGCG, larger, well-designed clinical trials are warranted to establish the efficacy and optimal dosage of EGCG in UF management.

EGCG exhibits the ability to impede ER signaling through the action of aromatase, an enzyme crucial in synthesizing estrogen. Consequently, this hinders estrogen-driven impacts on UFs cells, resulting in a decrease in their response to estrogen [160]. EGCG inhibits COMT, which eliminates the regulatory 2-hydroxy E2 to an abnormally elevated estrogen effect [144]. COMT enzyme, which is involved in estrogen metabolism, is overexpressed in UFs versus Myo-N. Moreover, higher amounts of COMT are observed in Black women compared to White [142]. EGCG bind to ERalpha and ERbeta and elicited ER-mediated gene expression in vitro and in vivo [161].

EGCG possesses immunomodulatory properties, which can help regulate and balance the immune system’s response. EGCG has been shown to inhibit the production of pro-inflammatory molecules, such as cytokines and chemokines, while promoting the production of anti-inflammatory substances [162]. Additionally, EGCG has been found to modulate immune cell function by enhancing the activity of natural killer (NK) cells, T cells, and B cells [163]. When pregnant C57BL/6 mice were exposed to 1 mg/kg body weight of EGCG dissolved in their drinking water from gestational days 0.5 to 16.5, the levels of proteins within the NF-κB and RAF/MEK/ERK signaling pathways exhibited significant elevation, potentially contributing to the onset of inflammation. The initiation of the TGFβ1)/Smad signaling pathway might play a role in the progression of endometrial fibrosis. Alterations in the expression of ER α and β (ERα, ERβ), progesterone receptor (PGR), and androgen receptor (AR) could potentially influence the aforementioned signaling pathways [164].

UF recurrence is a common concern after surgical removal or other treatment interventions [165]. EGCG, with these great properties, collectively contributes to inhibit UF growth and recurrence by targeting key pathways involved in UF development and progression [93,166].

EGCG has been explored for its potential role in reducing the severity of symptoms associated with UFs [167]. While research in this area is limited, some studies have suggested that EGCG may have beneficial effects on specific UF symptoms. It is believed that EGCG’s anti-inflammatory and antioxidant properties might contribute to symptom alleviation by reducing inflammation and oxidative stress, which are known to play a role in UF development and symptoms [168,169]. Moreover, it alleviates symptom severity in UFs and improves endometriosis through anti-fibrotic mechanisms. Additionally, it can reduce uterine contractility and improve the generalized hyperalgesia associated with dysmenorrhea and adenomyosis [170]. The results indicate that dietary supplementation with EGCG reduces the incidence and size of spontaneously occurring leiomyoma of the oviduct in Japanese quail (*Coturnix japonica*) [171]. 

EGCG exhibits several properties that have demonstrated potential in improving treatment outcomes for various conditions. For example, it has been explored for its ability to enhance the effectiveness of specific medications and therapies through synergistic interactions [168]. While further research is necessary to comprehensively grasp the underlying mechanisms and optimize treatment strategies, EGCG presents a promising option as an adjunct therapy to augment treatment outcomes in diverse health conditions. Epigenetic mechanisms, particularly the interplay between DNA methylation and gene expression, are the subject of intensive investigation in both malignant and benign tumors, including UFs. Integrating EGCG-based therapy into the treatment of women experiencing symptoms due to UFs may offer long-term benefits. Findings from in vitro and in vivo studies suggest the efficacy of EGCG treatment for UFs [172]. 

EGCG has been explored for its potential role in improving healthrelated QOL outcomes with positive effects on various aspects of well-being [173,174]. EGCG’s antioxidant and anti-inflammatory properties are believed to contribute to reduced oxidative stress and inflammation, which can improve overall health and QOL [24,175]. Moreover, EGCG has been associated with enhanced cognitive function, cardiovascular health, and weight management, all of which can impact one’s QOL [176] (Figure 2).

## 4. Summary of Supporting Clinical Trials

There are four registered clinical trials investigating the effects of EGCG and green tea catechins on UFs, with two trials in Italy (NCT05448365 and NCT05409872) and two in the United States. Three of these trials are currently recruiting participants to assess the impact of EGCG alone or in combination with other substances like vitamin B6 and D-Chiro-inositol. These trials aim to evaluate the effects on fertility and uterine fibroid volume.

One of the U.S. trials (NCT04177693) is specifically focused on the pharmacokinetics and hepatic safety of EGCG, examining various biochemical parameters. While more research is needed to directly link EGCG to QOL outcomes, it is suggested that incorporating EGCG-rich green tea into one’s lifestyle may have the potential to enhance overall well-being and health-related QOL. Furthermore, there’s an ongoing NICHD Confirm-funded trial called the FRIEND trial, which explores the use of EGCG in treating women with fibroids and unexplained infertility. Prior to this trial, a preliminary study known as pre-FRIEND (NCT04177693) was conducted to assess the safety of EGCG in premenopausal women [177].

In addition, clinical studies have explored the potential benefits of vitamin D and EGCG supplementation in managing UFs and have shown promising results. Several studies have indicated an inverse relationship between vitamin D levels and the risk of developing fibroids. For instance, an Iranian RCT administered 50,000 IU of vitamin D for 12 weeks, which inhibited UF growth compared to a placebo group. Additionally, combined supplementation of EGCG, vitamin D, and vitamin B6 has been found to reduce UF volume, alleviate symptoms, and improve the QOL in affected individuals.

Clinical trials investigating the safety and efficacy of EGCG as an anti-UFs treatment have reported significant decreases in UFs volume, improvements in UFs symptoms, and enhanced healthrelated QOL scores compared to control groups. However, further research with larger cohorts and randomized controlled trial designs is necessary to assess the clinical effects and potential side effects of vitamin D and EGCG supplementation, either alone or in combination, for managing UFs, especially in women approaching late reproductive life when hormonal manipulation may not be suitable.

It is important to note that relying solely on self-reported UF diagnoses or short-term vitamin D usage may not yield significant correlations. Therefore, raising awareness and prompting further investigation into watchful interventions is crucial. Several studies consistently demonstrated that vitamin D supplementation either stabilized or reduced UF size. The duration of treatment varied across studies, ranging from 8 to 16 weeks. Notably, vitamin D supplementation appeared safe within the prescribed dosages [106,121].

In a prospective double-blind study, women with vitamin D deficiency received 50,000 IU of vitamin D every 2 weeks for 10 weeks. The treated group exhibited a significant increase in its levels and a noteworthy reduction in UF volume at the six-month follow-up compared to the control group. Overall, these findings suggest that vitamin D and EGCG supplementation may hold promise in managing UFs, but further research is needed to fully understand their clinical effects and safety profiles [99,107,178,179,180]. 

The concomitant administration of them represents a promising treatment in women of late reproductive life for which hormonal manipulation is not foreseen [128,181]. The utilization of self-reported UF diagnosis as the primary metric or short-term usage of vitamin D might have correlation [182,183,184]. Because of these kinds of findings, there is an importance to raise awareness within the population, prompting further investigation for watchful interventions (Figure 3).

## 5. Avoiding Phthalate Exposure

Multiple EDCs are associated with UF outcomes and/or processes including phthalates, parabens, environmental phenols, alternate plasticizers, Diethylstilbestrol, organophosphate esters, and tributyltin [185,186,187,188]. Endocrine disruptions induced by environmental toxicants have placed an immense burden on society to properly diagnose, treat and attempt to alleviate symptoms and disease [189]. We have shown that the reprogramming of MSCs caused by early-life exposure to EDCs plays a significant role in UF development [190,191]. In addition, Bisphenol A can regulate integrin subunit alpha 2 (ITGA2) through the transcription factor XBP1, activate the downstream PI3K/AKT signaling pathway, and eventually promote the proliferation of UFs [192]. Emerging evidence suggests that phthalates can have long-term effects on reproductive health. It is important to adopt preventive measures to minimize exposure to these harmful substances [193,194]. Phthalates are commonly used to enhance the flexibility and durability of plastics, but their potential link to gynecological disorders, including UFs, requires further research [195,196]. Unfortunately, phthalates can still be found in various consumer products. Regulatory measures have been implemented to address their use, but individuals can also take proactive steps to minimize exposure [197,198]. The Eker rat model, the only authentic UF animal model, was used to assess the potential effects of early life exposure to EDCs and we identified an epigenetic mechanism of MLL1/HDAC-mediated MMSC reprogramming. EDC exposure epigenetically targets MMSCs and imparts an inflammatory responsive gene (IRG) expression pattern, which may result in a “hyper-inflammatory phenotype” and an increased hormone-dependent risk of UFs later in life [199,200]. Using this model, several in vitro assays were employed and demonstrated that compounds from three major classes of xenoestrogens can mimic the effect of estrogen on leiomyoma cells and act as estrogen receptor (ER) agonists. In addition, our data revealed that early-life developmental EDC exposure alters these MSCs’ DNA repairability and reverses induced DNA damage, providing a driver for the acquisition of mutations that may promote the development of these tumors later in adult life [191].

Evidence within the literature suggests that the body burden of environmental contaminants, especially in combination with inherent genetic variations, likely contributes to previously observed racial disparities in women’s health conditions such as breast cancer, endometriosis, polycystic ovarian syndrome, UFs, and premature birth [201,202]. In the cohort of Black women, exposure to phenols, parabens, and triclocarban was prevalent and several factors were associated with biomarker concentrations due to low socioeconomic level [203]. Recommendations should be made to reduce human exposure to EDCs and to protect from steadily increasing reproductive health risks [204,205]. 

Here are some simple guidelines to shield against phthalate exposure: Use phthalate-free water bottles.Stay hydrated with filtered water.Include detox foods in the diet.Adopt a plant-based diet intake.Consume fruits with vitamins and fiber content [206,207].Remove shoes before entering the home.Avoid microwaving plastics.Reduce the use of beauty products.Avoid nail polish.Minimize pesticide exposure.Use trappers without plastics at home.Wash hands regularly.Dust and vacuum often with a HEPA-filtered vacuum.Choose fresh whole foods over processed ones.Avoid air fresheners.Remove the top layer of food before usage.Limit shellfish consumption.Reduce the use of plastic containers.Reduce the consumption of high-fat dairy products [208,209,210,211].

In an attempt to protect the next generation of daughters from UFs, it is important now to choose toys wisely, use prescribed baby care products, limit exposure during pregnancy and breastfeeding, avoid ready-packaged powdered milk in plastic cans, and participate in pregnancy prevention programs [212,213].

When outdoors, it is advisable to read labels, avoid getting stuck in traffic or wear phthalate-resistant masks, filter tap water, watch out for leaching from plastics, seek alternatives to plastic containers, and utilize web-based and digital health interventions for information and support. Future research directions include studying the mechanisms of action, conducting epidemiological studies, identifying biomarkers, exploring intervention strategies, and examining long-term and transgenerational effects of phthalates on UFs [214,215,216]. Di(2-ethylhexyl)phthalate (DEHP) (phthalate parent) promoted cell viability and anti-apoptotic protein expression and induced HIF-1α and COX-2 expression in human leiomyoma cells [217]. Mono(2-ethyl-5-hydroxyhexyl) phthalate (MEHHP), the principal DEHP metabolite promotes UF survival by activating the tryptophan-kynurenine-AHR pathway [218]. Surprisingly, AHR was significantly overexpressed in uterine and this overexpression was correlated with living in Tehran (capital of Iran), smoking, living near polycyclic aromatic hydrocarbon producing companies and eating grilled meat [219]. Air pollution represents a relevant hormonal disruptor that acts on key signaling pathways, resulting in tumor development and infertility [220]. UF Patients had significantly higher levels of total urinary mono-ethylhexyl phthalate (ΣMEHP), mono-n-butyl phthalate (MnBP), and monoethyl phthalate (MEP) [221]. By addressing these research areas, we can improve our understanding of the impact of phthalates and develop strategies for prevention and improved patient outcomes (Figure 4).

Intensive lifestyle changes based on research can significantly reduce the risk of UFs. To minimize the risk, individuals should promote digestive tract, liver, and kidney function through regular bowel movements and the use of beneficial herbs like milk thistle and dandelion root [222,223,224]. Maintaining a healthy body weight, engaging in regular physical activity, and considering the use of hormonal contraceptives are important preventive measures. Conversely, individuals should manage chronic psychological stress, engage in regular exercise, limit alcohol consumption, and adopt a healthy diet that is low in processed and refined foods while being rich in fruits, vegetables, and fiber. By implementing these lifestyle changes, individuals can potentially lower their risk of UFs and improve their reproductive health. Consulting healthcare professionals for personalized advice and support is recommended.

Clinical studies and observational data have significantly contributed to our understanding of UFs and have provided valuable insights into their prevalence, risk factors, symptoms, diagnostic approaches, treatment options, and long-term outcomes. Rigorous research methodologies, such as RCTs and cohort studies, have allowed researchers to evaluate the effectiveness of various interventions, medications, and surgical techniques for managing UFs. In addition, observational data have been instrumental in identifying patterns, trends, and associations between different factors and the development or progression of UFs. This evidence-based information derived from clinical studies and observational data plays a crucial role in guiding medical decision making, informing guidelines and protocols, and ultimately improving patient care and outcomes for individuals with UFs. All 306 UF-related clinical trials were summarized in Appendix A.

## 6. Anti-UF Diet

Following long-lasting RCT or a long cohort showing that nutrition plays a crucial role in the management of UFs, adopting a personalized dietary regimen that includes specific foods, supplements, and vitamins can help reduce the risk of UFs, prevent their further growth, and alleviate symptoms [225]. A low intake of fruit and green vegetables and pollutants ingested with food is linked to a higher risk of myoma formation [226,227]. This includes incorporating vegetables rich in fiber, such as leafy greens, cruciferous vegetables, carrots, sweet potatoes, and bell peppers, which provide essential nutrients and antioxidants that combat inflammation and support hormonal balance [228]. Based upon observational and epidemiological investigations [135], adding fruits like berries, citrus fruits, apples, pineapple, and kiwi to the diet contributes antioxidants, anti-inflammatory compounds, and enzymes that may reduce the risk of UF growth [229]. Omega-3 fatty acids found in fatty fish, flaxseed, chia seeds, walnuts, nuts, and seeds have anti-inflammatory properties that can modulate inflammation associated with UF development [230]. Incorporating fish and lean meats like skinless chicken and egg whites provides high-quality protein and essential nutrients [231]. By following a comprehensive approach that includes a UF-preventive diet along with exercise, stress management, and medical guidance, individuals can promote gynecological health and potentially reduce the risk of UFs. 

In contrast, there are certain food items that women are advised to decrease or eliminate intake as they may potentially worsen the condition. Red meat, particularly processed and unprocessed beef and pork, contains high levels of saturated fats and potentially harmful compounds that can promote inflammation and hormonal imbalances in the body [232]. While more research is needed to establish a definitive link between red meat and UFs, it is generally recommended to maintain a balanced and varied diet with moderate red meat intake [233]. High-fat dairy products, which can contain high levels of estrogen and contribute to inflammation, are also believed to have a negative influence on fibroids. Choosing low-fat or skim dairy products as part of a balanced diet may be beneficial for individuals with UFs [15]. Caffeine, found in coffee, tea, energy drinks, and some sodas, has conflicting evidence regarding its impact on UFs [234]. Some studies suggest an increased risk with high caffeine intake, as caffeine can affect estrogen levels [233]. However, individual responses to caffeine can vary, and reducing caffeine intake may help alleviate symptoms for some women. The relationship between alcohol consumption and UFs is not fully understood, but excessive alcohol intake can lead to hormonal imbalances and impair liver function, potentially impacting hormone levels and UF development or progression. Moderate alcohol consumption may not have significant adverse effects, but excessive or chronic intake can be detrimental to overall health. It is important for women with UFs to discuss alcohol consumption with their healthcare provider to make informed decisions [235,236,237].

The utilization of natural compounds in the prevention and management of UFs holds promise for improved gynecological health. Curcumin, a natural phenol found in turmeric, exhibits anti-inflammatory and antioxidant properties that may have therapeutic potential in managing UFs [238,239,240]. Flaxseed, rich in lignans, is believed to have anti-inflammatory and antioxidant effects that could benefit individuals with UFs [241]. Resveratrol, present in mulberries, peanuts, and grapes [242], shows promise as an anti-fibrotic therapy by reducing ECM-related proteins and inhibiting UF cell proliferation [243,244]. Berberine, derived from plants like *Scutellaria barbata*, selectively inhibits UF cell proliferation induced by estrogen and progesterone, making it a potential targeted therapeutic option [245,246]. Methyl jasmonate obtained from jasmine plants demonstrates anti-EZH2 activity, suggesting its potential as a UF treatment [247]. Quercetin, found in various fruits and vegetables, modulates UF cell migration and gene expression [248], while sulforaphane, abundant in cruciferous vegetables, inhibits fibrotic processes associated with UFs [249]. Fucoidans, derived from brown seaweeds, exhibit antioxidant and anti-inflammatory properties and may attenuate UF progression [250]. Indole-3-carbinol, commonly found in cruciferous vegetables, regulates ECM components and cellular proliferation in UFs [251]. Isoliquiritigenin, present in licorice family plants, inhibits UF cell proliferation and activates apoptotic pathways [252,253]. Anthocyanins, lycopene, and vitamin A found in fruits and vegetables may contribute to UF prevention, but further research is required [254]. Chinese herbal preparations like Gui Zhi Fu Ling and Nona Roguy show potential in UF management [255,256]. Genistein, derived from soybeans, induces novel cell death pathways in UF cells [257]. Minerals like selenium and magnesium may have therapeutic effects on UFs, while probiotics promote a healthy gut microbiome, potentially reducing UF risk [258]. Incorporating whole grains into the diet may help maintain hormonal balance and reduce UF risk. It is important to consult healthcare professionals for personalized guidance on the usage of these natural compounds for UFs.

## 7. Conclusions

Screening and prevention measures for UFs, which are the most prevalent type of pelvic tumor in women under 50, hold immense significance due to the array of symptoms they induce, affecting a woman’s QOL. Detection and management of UFs take on special importance for women with plans for conception or facing fertility challenges. Despite their typically non-cancerous nature, UFs can occasionally lead to complications like organ compression or degeneration, underscoring the urgency of prompt medical attention. Effective management of risk factors is pivotal in addressing UFs, empowering healthcare practitioners to institute preemptive strategies and customize treatment approaches.

The dissemination of knowledge regarding these risk factors heightens consciousness and facilitates well-informed health decisions. Ideally, options such as the utilization of vitamin D (4000 IU/day), EGCG (800 mg/day), and EDC-free products can be considered safe, efficacious, and economically viable for extended usage for women who have undergone myomectomy to prevent the recurrence of fibroids, as well as for those experiencing early symptoms and displaying findings on imaging. Additionally, our team is focused on the application of the suggested approach as a clinical trial named ERADICATE-UF. 

Strategies aimed at potentially mitigating the risk or impact of UFs encompass sustaining a healthy body weight, adhering to a nourishing diet, regulating hormone levels, and undergoing regular health assessments. Nonetheless, achieving complete prevention might be unattainable, underscoring the significance of personalized guidance from healthcare providers.

## Figures and Tables

**Figure 1 ijms-24-15972-f001:**
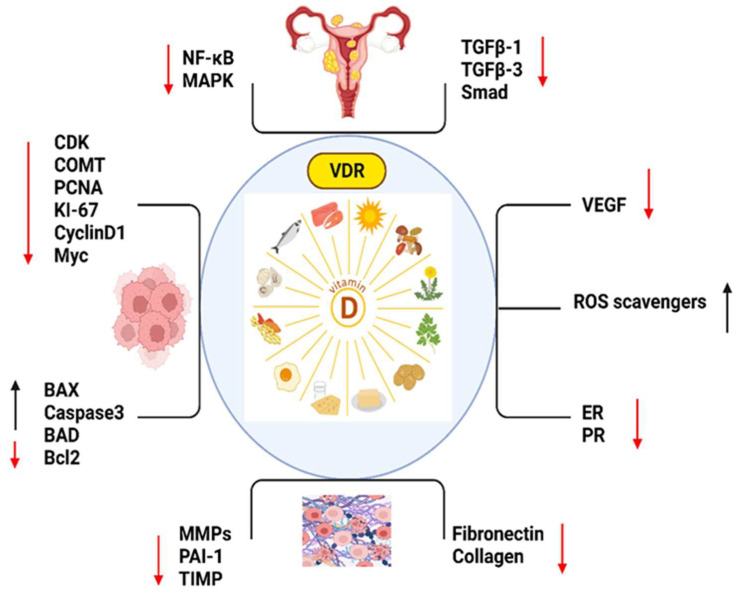
The important markers which play roles in vitamin D such as, NF-KB: Nuclear Factor-Kappa B; MAPK: Mitogen-Activated Protein Kinase; TGFB1: Transforming Growth Factor Beta 1; TGFB3: Transforming Growth Factor Beta 3; SMAD: Mothers Against Decapentaplegic (SMAD is a family of proteins); CDK: Cyclin-Dependent Kinase; COMT: Catechol-O-Methyltransferase; PCNA: Proliferating Cell Nuclear Antigen; KI-67: Antigen KI-67 (also known as MKI67); CYCLIN D1: Cyclin D1; MYC: Myelocytomatosis Oncogene; BAX: Bcl-2 Associated X Protein; CASPASE 3: Caspase 3; BAD: Bcl-2 Associated Death Promoter; BCL2: B-Cell Lymphoma 2; MMPS: Matrix Metalloproteinases (MMPS is a family of proteins); PAI-2: Plasminogen Activator Inhibitor-2; TIMP: Tissue Inhibitor of Metalloproteinases (TIMP is a family of proteins); FIBRONECTIN: Fibronectin; COLLAGEN: Collagen; VEGF: Vascular Endothelial Growth Factor; ROS SCAVENGERS: Reactive Oxygen Species Scavengers; ER: Estrogen Receptor and PR: Progesterone Receptor. (Red arrow: decreased expression, black arrow: increased expression.)

**Figure 2 ijms-24-15972-f002:**
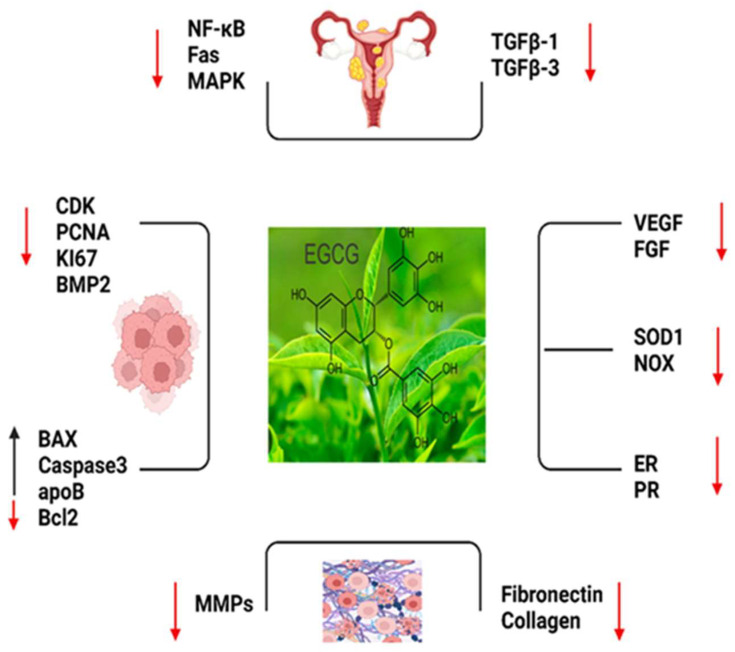
The important markers which play roles in EGCG such as, NF-KB: Nuclear Factor-Kappa B; Fas; Fas cell surface death receptor; MAPK: Mitogen-Activated Protein Kinase; TGFB1: Trans-forming Growth Factor Beta 1; TGFB3: Transforming Growth Factor Beta 3; CDK: Cy-clin-Dependent Kinase; PCNA: Proliferating Cell Nuclear Antigen; KI-67: Antigen KI-67 (also known as MKI67); BMP2: Bone Morphogenetic Protein 2; BAX: Bcl-2 Associated X Protein; CASPASE 3: Caspase 3; apoB: Apolipoprotein B; BCL2: B-Cell Lymphoma 2; MMPS: Matrix Metalloproteinases (MMPS is a family of proteins); FIBRONECTIN: Fibronectin; COLLAGEN: Collagen; VEGF: Vascular Endothelial Growth Factor; FGF: Fibroblast Growth Factor; SOD1: Superoxide Dismutase 1; NOX: NADPH Oxidase; ER: Estrogen Receptor and PR: Progesterone Receptor (Red arrow: decreased expression, black arrow: increased expression.).

**Figure 3 ijms-24-15972-f003:**
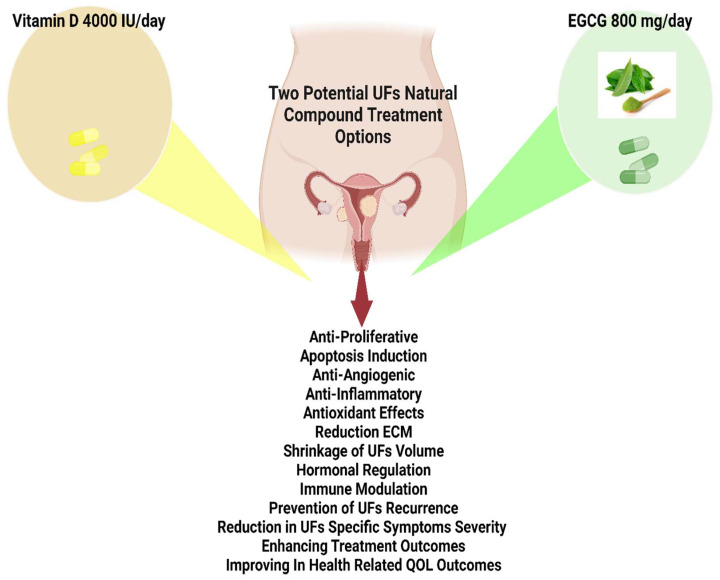
Utilization of vitamin D (4000 IU/day), and epigallocatechin gallate (EGCG), (800 mg/day) can be considered safe and efficacious for extended usage based on related studies.

**Figure 4 ijms-24-15972-f004:**
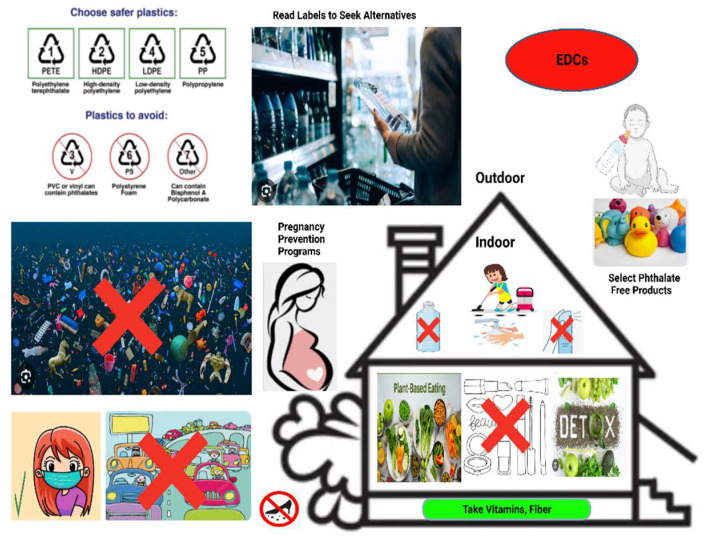
Simple but important considerations to avoid endocrine-disrupting chemical (EDC)-containing products, such as using phthalate-free products, reading labels to seek alternatives, reducing the use of plastic containers, avoiding traffics jam, removing shoes before entering the home, reducing the use of beauty and cosmetic products like nail polish, avoiding water plastic bottle and air freshener, staying hydrated with filtered water, wearing mask, including detox foods, adopting a plant-based diet intake, consuming fruits with vitamins and fiber content, dusting and vacuuming often with a high-efficiency particulate absorbing filter (HEPA)-filtered vacuum, choosing fresh whole foods over processed ones.

## Data Availability

Not applicable.

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
