# Peer review of "Evidence-Based Approach for Secondary Prevention of Uterine Fibroids (The ESCAPE Approach)"

_ijms, 2023, doi:10.3390/ijms242115972_

Round 1

Reviewer 1 Report

Comments and Suggestions for Authors

Article Review: "Evidence-Based Approach for Secondary Prevention of Uterine Fibroids (The ESCAPE Approach)"

Summary: This literature review examines the use of dietary supplements, diet, and lifestyle changes for the prevention of uterine fibroids. The authors present evidence supporting the proposed comprehensive approach, known as the ESCAPE approach.

Comments/Revisions:

1.       The authors describe their strategy as an approach for the secondary prevention of uterine fibroids. Given that the proposed interventions are cost-effective and low in toxicity, and considering the significant socioeconomic implications of fibroids, it is worth exploring why this approach doesn't focus on primary prevention. Are there concerns regarding the use of these proposed methods by asymptomatic women without abnormalities detected through imaging?

2.       If the authors feel that secondary prevention is the primary objective of their intervention, they should specify the target group. Is this approach recommended for women who have undergone myomectomy to prevent recurrence of the fibroids, those experiencing early symptoms and showing findings on imaging, or both? Please elaborate on this point in the 'Discussion' section.

3.       Could these interventions be effectively combined with hormonal modalities? Explore the potential synergy or interaction between the proposed interventions and hormonal treatments.

4.       Is there a plan to conduct clinical trials to evaluate the proposed approach? Please elaborate in the 'Discussion' section.

Recommendation: Minor revision

Comments on the Quality of English Language

Article Review: "Evidence-Based Approach for Secondary Prevention of Uterine Fibroids (The ESCAPE Approach)"

Summary: This literature review examines the use of dietary supplements, diet, and lifestyle changes for the prevention of uterine fibroids. The authors present evidence supporting the proposed comprehensive approach, known as the ESCAPE approach.

Comments/Revisions:

1.       The authors describe their strategy as an approach for the secondary prevention of uterine fibroids. Given that the proposed interventions are cost-effective and low in toxicity, and considering the significant socioeconomic implications of fibroids, it is worth exploring why this approach doesn't focus on primary prevention. Are there concerns regarding the use of these proposed methods by asymptomatic women without abnormalities detected through imaging?

2.       If the authors feel that secondary prevention is the primary objective of their intervention, they should specify the target group. Is this approach recommended for women who have undergone myomectomy to prevent recurrence of the fibroids, those experiencing early symptoms and showing findings on imaging, or both? Please elaborate on this point in the 'Discussion' section.

3.       Could these interventions be effectively combined with hormonal modalities? Explore the potential synergy or interaction between the proposed interventions and hormonal treatments.

4.       Is there a plan to conduct clinical trials to evaluate the proposed approach? Please elaborate in the 'Discussion' section.

Recommendation: Minor revision

Author Response

Reviewer 1

Summary: This literature review examines the use of dietary supplements, diet, and lifestyle changes for the prevention of uterine fibroids. The authors present evidence supporting the proposed comprehensive approach, known as the ESCAPE approach.

Comments/Revisions:

  1. The authors describe their strategy as an approach for the secondary prevention of uterine fibroids. Given that the proposed interventions are cost-effective and low in toxicity, and considering the significant socioeconomic implications of fibroids, it is worth exploring why this approach doesn't focus on primary prevention. Are there concerns regarding the use of these proposed methods by asymptomatic women without abnormalities detected through imaging?

Response:

We appreciate your insightful question regarding applying our approach for the primary prevention of uterine fibroids. We are actually exploring that notion experimentally since there is currently no conclusive evidence in the literature to support the use of Vitamin D (4000IU/day) and EGCG (800mg/day) for primary prevention of uterine fibroids (UFs) to slow/prevent tumor appearance. Primary prevention aims to stop the development of a disease in healthy individuals before it begins, but UFs lack a reliable screening method, making it challenging to identify at-risk individuals early. Uterine fibroids are typically diagnosed through imaging techniques when symptoms are present, and therefore, the strategy discussed here primarily falls under secondary prevention. It focuses on early detection and addressing UFs in individuals who exhibit subtle signs of illness, particularly those concerned about recurrence after surgery. There is no validated method for screening asymptomatic women for their risk of future UFs which is worth further investigation. This secondary prevention approach aims to prevent UFs' progression or recurrence, offering safe and fertility-friendly options to potentially reduce the global prevalence of the condition. While primary prevention is a noble goal, addressing the complexities of UFs requires a multifaceted approach that includes secondary prevention strategies like the one proposed.

  1. If the authors feel that secondary prevention is the primary objective of their intervention, they should specify the target group. Is this approach recommended for women who have undergone myomectomy to prevent recurrence of the fibroids, those experiencing early symptoms and showing findings on imaging, or both? Please elaborate on this point in the 'Discussion' section.

Response:

We thank you for your perceptive question about the suitability of this approach for women who have undergone myomectomy to prevent the recurrence of fibroids, as well as for those experiencing early symptoms and displaying findings on imaging. Both groups of these women can empower themselves to manage their health and mitigate the risks associated with UFs by adhering to the provided guidelines and we added it in conclusion part, line 746-748.

  1. Could these interventions be effectively combined with hormonal modalities? Explore the potential synergy or interaction between the proposed interventions and hormonal treatments.

Response:

We're grateful for your insightful feedback. Combining Vitamin D (4000IU/day) and EGCG (800mg/day) interventions with hormonal treatments for uterine fibroids management is a promising avenue for future research. Especially that hormonal therapies offer symptom relief but they come with drawbacks like weight gain and mood swings, making them unsuitable for all patients for limited duration only which makes our proposed natural intervention suitable to serve as long term option.

The suggested secondary prevention approach, exemplified by the ESCAPE (Evidence-Based Approach for Secondary Prevention) for uterine fibroids management, focuses on natural and safe interventions to alleviate patient symptoms, reducing reliance on invasive surgical and pharmacological treatments for advanced disease. This cost-effective strategy aims to enhance patients' quality of life by addressing early disease, slowing its progression, or potentially stopping it altogether. Traditionally, medical intervention for uterine fibroids has been delayed until the disease burden is high, leading to surgical procedures. However, this approach is linked to high recurrence rates, necessitating additional surgeries and interventions, which are both invasive and costly and have negative effects on women's health. The ESCAPE approach presents a promising non hormonal alternative by incorporating straightforward, cost-effective, and safe measures that could prevent the development of uterine fibroids, promote overall reproductive health, reduce the need for unnecessary surgeries, and save substantial healthcare resources globally. This proactive secondary prevention strategy signifies a crucial shift in uterine fibroid management.

  1. Is there a plan to conduct clinical trials to evaluate the proposed approach? Please elaborate in the 'Discussion' section.

Response:

We are so grateful to your great suggestions, we are going to assessing the suggested approach, we are in the process of planning a clinical trial named ERADICATE-UF. This trial aims to investigate the impact of a combination of Vitamin D and EGCG on the secondary prevention of uterine fibroids in women of color. Currently, it is undergoing review by the Institutional Review Board (IRB). According to your great comment, we added it in conclusion part, line 748-749.

Reviewer 2 Report

Comments and Suggestions for Authors

Dear author’s 

I was pleased to review your article and i have the following comment’s:

First of all the different methods to prevent uterine fibroids development are experimental and there are no guideline’s with this recommendation. So this different substances or alimentary supplements are not recommend for the growing myoma  prevention.

A section methodology is informative in order to explain the data collection and review.

Prospective studies should demonstrate if they are some molecules with the role of growing fibroma inhibition.

Please explain what new info brings your study in the actual literature.

Author Response

Reviewer 2

Dear author’s 

I was pleased to review your article and I have the following comments:

  1. First, the different methods to prevent uterine fibroids development are experimental and there are no guidelines with this recommendation. So, these different substances or alimentary supplements are not recommended for the growing myoma prevention.

Respond:

Your excellent comment is much appreciated. Currently, there are ongoing efforts to explore methods for preventing the development of uterine fibroids. In our manuscript, we have delved into the available data in the literature and categorized our findings into three key areas. In Part 2, we focused on Vitamin D serum level monitoring and appropriate supplementation. Although established guidelines supporting the use of Vitamin D for fibroid prevention are lacking, our review of the literature suggests potential benefits, and there are clinical trials in Part 4 that further support this avenue of slowing tumor growth compared to control group. In Part 3, we examined Epigallocatechin gallate (EGCG) and green tea extract consumption. While there are no firmly established guidelines for using EGCG and green tea extracts to prevent fibroids, we have identified relevant clinical trials in Part 4 that lend credence to our recommendation. It's important to note that these various substances and dietary supplements are not currently mainstream recommendations for fibroid prevention. Therefore, it is crucial to base decisions on evidence-based medical guidance when considering preventive measures for uterine fibroids. Consultation with a healthcare provider is strongly advised to explore the most appropriate and effective strategies in this context. The ongoing clinical trials highlighted in Part 4 contribute to the evolving understanding of these approaches and may offer further insight into their potential for fibroid prevention.

  1. A section methodology is informative to explain the data collection and review.

Respond:

Your thoughtful comment is truly valued. Within our manuscript's methodology, we undertook a comprehensive search for studies across multiple databases, including PubMed, Scopus, Embase, and ISI Web of Science related to the early detection and prevention of uterine fibroids, aiming to help women with strategies to reduce their risk of developing uterine fibroids and improve their overall gynecological health. According to your great comment, we added it in line 115-119.

  1. Prospective studies should demonstrate if they are some molecules with the role of growing fibroma inhibition.

Respond:

We want to express our gratitude for your comment. Your suggestion regarding prospective studies exploring molecules for inhibiting fibroma growth is greatly appreciated. However, it's crucial to emphasize that our manuscript does not involve conducting original prospective studies. Rather, it constitutes an evidence-based review article where our goal is to offer a thorough analysis of the existing evidence related to the topic. Your input is valuable, and we concur that previous and future prospective studies could substantially enhance our understanding of the role of specific molecules in inhibiting fibroma growth.

  1. Please explain what new info brings your study in the actual literature.

Respond:

We thank the reviewer for his/her comment. Our manuscript represents a comprehensive review article that goes beyond the surface of existing literature to identify gaps and shed light on the potential applications of Vitamin D (4000IU/day) and EGCG (800mg/day) in the realm of secondary prevention for uterine fibroids. We offer pragmatic, non-invasive, and secure perspectives on this prevalent health concern.

Within the manuscript, we introduce a groundbreaking approach known as ESCAPE (Evidence-Based Approach to Secondary Prevention of UFs). This innovative strategy is designed to address uterine fibroids through a series of straightforward, cost-effective, and safe interventions. By implementing the ESCAPE approach, we aim to not only halt the progression of uterine fibroids but also enhance overall reproductive health. We anticipate a reduction in the number of unnecessary surgical procedures, resulting in substantial cost savings for healthcare systems worldwide, potentially amounting to billions of dollars.

A focus of the ESCAPE approach is the emphasis on early detection and the efficient management of uterine fibroids. We have tried incorporating these critical aspects into both sections of the abstract and conclusion of our manuscript, ensuring that the reader gains a holistic understanding of the multifaceted approach we propose. Through our work, we aspire to catalyze a paradigm shift in the management of uterine fibroids, offering a promising path towards better health outcomes for women.

Round 2

Reviewer 2 Report

Comments and Suggestions for Authors

Thank you for your response.

Author Response

Thanks so much for your review